# Human Lupus Plasma Pro-Atherogenic Effects on Cultured Macrophages Are Not Mitigated by Statin Therapy: A Mechanistic LAPS Substudy

**DOI:** 10.3390/medicina55090514

**Published:** 2019-08-21

**Authors:** Allison B. Reiss, Hirra A. Arain, Lora J. Kasselman, Heather A. Renna, Juan Zhen, Iryna Voloshyna, Joshua DeLeon, Steven E. Carsons, Michelle Petri

**Affiliations:** 1NYU Winthrop Biomedical Research Institute and NYU Long Island School of Medicine, Mineola, NY 11501, USA; 2Division of Rheumatology, The Johns Hopkins University School of Medicine Baltimore, MD 21205, USA

**Keywords:** lupus, atherosclerosis, cholesterol transport, macrophage, statin

## Abstract

*Background and Objectives:* Atherosclerotic cardiovascular disease (CVD) remains a major cause of morbidity and mortality in persons with systemic lupus erythematosus (SLE, lupus). Atherosclerosis, which involves interplay between cholesterol metabolism and cellular inflammatory pathways, is primarily treated with statins since statins have lipid-lowering and anti-inflammatory properties. The Lupus Atherosclerosis Prevention Study (LAPS) was designed to investigate the efficacy of statins against CVD in SLE patients. LAPS demonstrated that 2 years of atorvastatin administration did not reduce atherosclerosis progression in lupus patients. In this LAPs substudy, we use cultured macrophages to explore the atherogenic properties of plasma from LAPS subjects to explain the mechanistic rationale for the inability of statins to reduce CVD in lupus. *Materials and Methods*: THP-1 differentiated macrophages were treated for 18 h with 10% SLE patient plasma obtained pre- and post-atorvastatin therapy or placebo. Gene expression of the following cholesterol transport genes was measured by qRT-PCR. For efflux—ATP binding cassette transporter (ABC)A1 and ABCG1, 27-hydroxylase, peroxisome proliferator-activated receptor (PPAR)γ, and liver X receptor (LXR)α; and for influx—cluster of differentiation 36 (CD36) and scavenger receptor (ScR)A1. *Results:* Macrophages exposed to plasma from both statin-treated and placebo-treated groups showed a significant decrease in cholesterol efflux proteins ATP binding cassette (ABC) transporters A1 and ABCG1, an increase in 27-hydroxylase, an increase in the LDL receptor and a decrease in intracellular free cholesterol. No change in influx receptors ScRA1 and CD36, nor nuclear proteins LXRα and PPARγ was observed. *Conclusions:* Statins do not normalize pro-atherogenic changes induced by lupus and these changes continue to worsen over time. This study provides mechanistic insight into LAPS findings by demonstrating that statins are overall ineffective in altering the balance of cholesterol transport gene expression in human macrophages. Furthermore, our study suggests that statins as a CVD treatment may not be useful in attenuating lipid overload in the SLE environment.

## 1. Introduction

Systemic lupus erythematosus (SLE) is a chronic multisystem autoimmune disorder associated with increased mortality. The risk of cardiovascular disease (CVD) in patients with SLE is inordinately high, and despite improved SLE treatment efficacy, incidence of myocardial infarction in young women is elevated and overall cardiovascular mortality is little changed in 30 years [1,2]. The pathogenic mechanisms underlying early CVD in SLE are complex and involve more than traditional CVD risk factors. SLE-associated factors that may contribute are likely related to immune dysregulation and inflammation [3].

Statin treatment to reduce low density lipoprotein (LDL) cholesterol is associated with the prevention of cardiovascular events. Statins have been the cornerstone of first line therapy for prevention and treatment of CVD for the last 30 years [4,5]. Despite maximally tolerated statin therapy, the residual risk of cardiovascular events remains significant [6,7] and this is a major problem in statin-treated SLE patients. The Lupus Atherosclerosis Prevention Study [8] (LAPS) was a randomized, double-blind, placebo-controlled trial of atorvastatin 40 mg versus matching placebo in 200 patients without clinical CVD enrolled from the Hopkins Lupus Cohort. In this group (92% female, 61% Caucasian, 34% African-American, 2% Asian, and 2% Hispanic, mean age 44 ± 11 years), the trial showed that after 2 years, atorvastatin failed to affect atherosclerotic disease progression as measured by coronary artery calcium (assessed by multidetector computed tomography) and carotid intima-media thickness measurement (by carotid duplex ultrasound).

Analysis of pro- and anti-atherogenic factors present in plasma from patients with autoimmune diseases, including SLE, can be analyzed in a macrophage cell culture system [9,10]. This in vitro assay consists of exposure of naïve cultured cells of the type found in the arterial wall (THP-1 monocytes/macrophages, endothelium) to plasma from patients, followed by gene and protein expression analysis. In this way, our group has documented atherogenic effects of plasma obtained from patients with SLE on cholesterol transport genes (Figure 1).

SLE plasma exhibits atherogenic potency by markedly stimulating the CD36 scavenger receptor and suppressing the reverse cholesterol transport protein 27-hydroxylase in cultured THP-1 monocytes as compared to healthy control plasma [9]. SLE plasma suppresses ATP binding cassette (ABC) A1 and ABCG1 expression in THP-1 macrophages by about 60% and 50%, respectively [11]. SLE plasma also reduces 27-hydroxylase expression in human aortic endothelial cells by about 50% [12].

In this LAPS substudy, we examined the atherogenicity of lupus plasma obtained from study subjects before and after atorvastatin or placebo therapy. In support of the parent study, we found that plasma from the same person post-therapy versus pre-therapy failed to improve lipid transport pathways or cholesterol efflux in naïve macrophages. However, we did observe changes in intracellular cholesterol level, as expected based on the properties of statins as inhibitors of hydroxymethylglutaryl CoA reductase (HMG-CoA). Therefore, we propose that statins are not sufficient to correct lupus-related abnormalities in cholesterol management at the cellular level because they do not target lupus-specific pro-atherogenic mechanisms and therefore, are inadequate for CVD prevention in this unique high CVD risk patient population.

## 2. Materials and Methods

### 2.1. Subject Inclusion and Exclusion Criteria

LAPS, the parent study, was approved by The Johns Hopkins Medicine institutional review board. The research was carried out according to The Code of Ethics of the World Medical Association, Declaration of Helsinki (application number 00-09-14-02, approval date 03/11/04) and informed consent was obtained. Patients included in the trial met all of the following criteria:
Patients with a clinical diagnosis of SLE, confirmed by a faculty rheumatologist at Johns Hopkins.Age 18 years or older.Given informed consent.


Patients were excluded from the trial if they met any one of the following exclusion criteria:SLE patients with a known atherosclerotic event, such as angina, myocardial infarction, or stroke, with an abnormal lipid profile for which a statin would be standard of care.Pregnant patients (or patients planning to become pregnant in the next two years).Patients with known chronic liver disease, unexplained elevation of their liver enzymes greater than 2-times the upper limit of normal, or an elevated creatine phosphokinase greater than 1.5-times the upper limit of the normal value for the patient’s racial group.Patients with triglycerides >500 mg/dl or LDL >190 in the absence of two risk factors (and who were unwilling to participate in a formal nutritional/lifestyle modification program that was recommended for them).

### 2.2. Statin or Placebo Treatment

The current experiment is a substudy of LAPS in which lupus patients were treated with 40 mg atorvastatin or placebo therapy; further details can be found in Petri et al., 2011 [8].

### 2.3. Cell Culture and Experimental Conditions

THP-1 monocytes (1 × 10^6^ cells/mL) were grown at 37 °C in a 5% CO_2_ atmosphere in RPMI 1640 as described [10]. To facilitate differentiation into macrophages, THP-1 monocytes were treated with 100 nM phorbol 12-myristate 13-acetate (PMA) for 24 h at 37 °C. Upon PMA removal, unattached cells in the supernatant were counted so that percentage adherence could be calculated. Attachment was uniform and exceeded 90%. The macrophages were equilibrated with RPMI 1640 growth medium without PMA for 24 h to allow cell recovery. Cell differentiation, morphology, and uniformity was verified by evaluating cell adhesion and spreading under an optical microscope.

THP-1 macrophages were then incubated in 12-well plates for 18 h in RPMI medium containing each of the following: (a) 10% fetal bovine serum (control); (b) 10% plasma from each SLE subject prior to placebo therapy; (c) 10% plasma from each SLE patient after placebo therapy; (d) 10% plasma from each SLE subject prior to statin therapy; (e) 10% plasma from each SLE patient after statin therapy. No fetal bovine serum was added to cells incubated with human plasma.

### 2.4. RNA Isolation and Gene Expression Analysis by QRT-PCR

RNA was isolated using 1 mL Trizol reagent per 10^6^ cells and dissolved in nuclease-free water. The quantity of total RNA from each condition was measured by absorption at 260 and 280 wavelengths by ultraviolet spectrophotometry (Hitachi U2010 spectrophotometer). ABCA1 [13], ABCG1 [14], 27-hydroxylase [15], liver x receptor (LXR)α [14], peroxisome proliferator-activated receptor (PPAR)γ [16], scavenger receptor (ScR)A1 [17], lectin-like oxidized low-density lipoprotein receptor -1 (LOX)-1 [17], CD36 [17], and native LDL receptor [18] were measured using QRT-PCR analysis with the FastStart SYBR Green Reagents Kit according to the manufacturers’ instructions on the Roche Light Cycler 480 (Roche Applied Science, Indianapolis, IN, USA). cDNA was copied from 1 µg of total RNA using Murine Leukemia Virus reverse transcriptase primed with oligo dT. Equal amounts of cDNA were taken from each reverse transcription reaction mixture for real-time PCR amplification using gene specific primers. Using the 2ΔΔCt method, the Ct value for each gene was normalized to that of glyceraldehydes-3-phosphate dehydrogenase (GAPDH), and then relative expression level was calculated as the mean value of the THP-1 cells treated with 10% plasma before statin therapy group as 1.0. Non-template controls were included for each primer pair to check for significant levels of any contaminants. A melting-curve analysis was performed to assess the specificity of the amplified PCR products. All PCR results presented below use as a baseline (set at 1.0) mRNA and protein levels in THP-1 macrophages exposed to 10% plasma before statin treatment. Plasma from each individual subject obtained before and after statin therapy was run in triplicate and the mean of these three values was used for analysis. The data represent the results from before and after statin pairs of subjects, presented as fold-change ± 95% confidence interval (CI).

### 2.5. Cholesterol Efflux Analysis

Cholesterol efflux was analyzed on THP-1 cells plated in 96 well plates at 1 × 10^6^ cells/mL in the presence of 10% pre- and post-statin and placebo-treated plasma using the Amplex Red Cholesterol Assay kit (Molecular Probes, Eugene, OR, USA), according to the manufacturer’s protocol. Performing reactions in the presence and absence of cholesterol esterase, total (TC) and free (FC) cholesterol were analyzed. Cholesterol esters (CE) were estimated as the difference between TC and FC and the CE/FC ratio was calculated. Fluorescence was read at 585 nm and cholesterol efflux was expressed as percentage of fluorescence in efflux medium to total fluorescence of the cells and medium combined.

### 2.6. Oxidized Cholesterol Uptake Analysis

Oxidized LDL (oxLDL) uptake was analyzed on THP-1 cells plated in 8 well chamber slides at 250,000 cells/mL in the presence of 10% plasma taken from subjects pre- and post-statin or pre- and post-placebotreatment for 18 h then incubated with 0.25 µg/mL 1,1’-dioctadecyl-3,3,3’,3’-tetramethylin docarbocyaninet (DiI)-oxLDL (Intracel, Frederick, MD, USA) for 4 h. Cells were fixed in 4% paraformaldehyde, washed with sterile PBS, and then prepared using Vectashield mounting medium containing DAPI stain (Vector Laboratories, Inc., Burlingame, CA, USA). After incubation, accumulation of DiI-oxLDL in cells was determined by fluorescent intensity with a Nikon A1 microscopy unit with 20× magnification and photographed with a DS-Ri1 digital camera. Fluorescent intensity was quantified from 9 random fields (1024 × 1024 pixels) per slide and maximum corrected total cell fluorescence (FU) was calculated [19].

### 2.7. Data Analysis

Statistical analysis was performed using Graphpad Prism, version 6 (GraphPad Software, San Diego, CA, USA). All normally distributed data were analyzed by mixed factorial analysis of variance with time (before and after therapy) as the repeated measure. Appropriate non-parametric equivalents were used for non-normal data. For PCR data, average dCt values were used for analysis and data are presented as fold change ± 95% CI. Probability values less than 0.05 were regarded as significant.

## 3. Results

### 3.1. Demographics, Clinical Characteristics, and Immunological Status of SLE Patients

There were 44 participant pre- and post-treatment plasmas (n = 21 placebo-treated and n = 23 atorvastatin-treated) included in this LAPS substudy. The placebo-treated substudy participants were 100% female, 0% male; 48% African American; 52% Caucasian; 32 ± 12 years of age. Placebo-treated substudy participants’ clinical and immunological characteristics collected at entry were as follows: disease duration 0–20 years; 100% with presence of anti-ANA antibodies; 67% with presence of anti-dsDNA antibodies; 10% with presence of anti-Sm antibodies; 30% with anti-RNP antibodies; 69% with low C3 or C4; and 17% with lupus nephritis (Table 1). The atorvastatin-treated substudy participants were 87% female, 13% male; 65% African American; 35% Caucasian; 34 ± 10 years old. Atorvastatin-treated substudy participants’ clinical and immunological characteristics collected at entry were as follows: disease duration 0–22 years; 100% with presence of anti-ANA antibodies; 74% with presence of anti-dsDNA antibodies; 18% with presence of anti-Sm antibodies; 30% with anti-RNP antibodies; 67% with low C3 or C4; and 32% with lupus nephritis (Table 1).

### 3.2. Comparison of Nuclear Receptor Expression (PPARγ and LXRα) in THP-1 Macrophages Exposed to SLE Plasma Obtained Pre- and Post-Placebo or Statin Therapy

In all experiments, mRNA level for the protein of interest in pre-placebo or pre-atorvastatin plasma-treated cells were set at 1.0.

PPARγ mRNA level in macrophages exposed to 10% plasma obtained from SLE subjects post-placebo treatment and 10% plasma obtained from SLE subjects post-atorvastatin-treatment showed no change at 1.13 (95% CI (0.67 to 1.90)) and 0.99 (95% CI (0.70 to 1.42), F_interaction_ (1, 29) = 0.206, *p* = 0.65; Figure 2) versus cells exposed to pre-placebo- or pre-atorvastatin plasma.

Macrophages exposed to 10% plasma obtained post-placebo-treatment and post-atorvastatin-treatment display no change in LXRα mRNA at 1.04 (95% CI (0.78 to 1.43)) and 1.20 (95% CI (0.844 to 1.71), F_time_ (1, 29) = 2.251, *p* = 0.1444; Figure 2), respectively versus cells exposed to respective pre-treatment plasma.

### 3.3. Comparison of mRNA Level of Reverse Cholesterol Transport Proteins in THP-1 Macrophages Exposed to SLE Plasma Obtained Pre- and Post—Placebo or Statin Therapy

In all experiments, mRNA levels of the protein of interest in macrophages exposed to pre-treatment plasma was set at 1.0.

THP-1 macrophages exposed to 10% plasma obtained post-placebo treatment display decreased ABCA1 mRNA to 0.48 (95% CI (0.29 to 0.82)) versus cells exposed to respective pre-treatment plasma (Figure 3). The same response we have observed in macrophages exposed to 10% plasma obtained post-atorvastatin treatment 0.48 (95% CI (0.24 to 0.94), F_time_ (1, 47) = 27.37, *p* < 0.0001).

Similarly, we observed a decrease in ABCG1 mRNA level when macrophages were exposed to 10% plasma obtained post-placebo- and post-atorvastatin-treatments. Thus, fold change in ABCG1 expression decreased to 0.47 (95% CI (0.29 to 0.75)) and to 0.49 (95% CI (0.29 to 0.80), F_time_ (1, 47) = 15.68, *p* < 0.0001), respectively.

The opposite change was detected in the expression of 27-hydroxylase. Thus, macrophages exposed to 10% plasma obtained post-placebo treatment display increased 27-hydroxylase mRNA to 1.69 (95% CI (1.18 to 2.41)), while macrophages exposed to 10% plasma obtained post-atorvastatin-treatment display increased 27-hydroxylase mRNA to 1.77 (95% CI (1.19 to 2.64), F_time_ (1, 32) = 50.28, *p* < 0.0001; Figure 3) versus cells exposed to respective pre-treatment plasma set at 1.0.

### 3.4. Comparison of mRNA Level for Scavenger Receptors and Ldl- Receptor Message in THP-1 Macrophages Exposed to Sle Plasma Obtained Pre- and Post-Placebo or Statin Therapy

In all our experiments mRNA levels for the protein of interest in pre-placebo or pre-atorvastatin plasma-treated cells were set at 1.0.

CD36 mRNA in macrophages exposed to 10% plasma obtained from SLE subjects post-placebo-treatment trended towards decreased expression at 0.59 (95% CI (0.33 to 1.06)), while macrophages exposed to 10% plasma obtained from SLE subjects post-atorvastatin treatment display no change in CD36 mRNA at 1.08 (95% CI (0.62 to 1.87), F_interaction_ (1, 47) = 2.802, *p* = 0.100; Figure 4) compared to cells exposed to respective pre-treatment plasma set at 1.0. ScRA1 mRNA in macrophages exposed to 10% plasma obtained from SLE subjects post-placebo treatment and 10% plasma obtained from SLE subjects post-atorvastatin-treatment showed no change at 0.84 (95% CI (0.46 to 1.54)) and 0.90 (95% CI (0.42 to 1.91), F_interaction_ (1, 32) = 0.062, *p* = 0.81; Figure 4) compared to cells exposed to respective pre-treatment plasma set at 1.0. LOX1 mRNA in macrophages exposed to 10% plasma obtained from SLE subjects post-placebo-treatment and 10% plasma obtained from SLE subjects post-atorvastatin treatment showed no change at 0.92 (95% CI(0.64 to 1.34)) and 1.07 (95% CI (0.69 to 1.68), F_interaction_ (1, 32) = 0.414, *p* = 0.53; Figure 4) compared to cells exposed to respective pre-treatment plasma set at 1.0.

THP-1 macrophages exposed to 10% plasma from subjects pre- and post-statin therapy both display a trend towards increased LDL receptor mRNA at 1.35 (95% CI (0.99 to 1.85), F_treatment_ (1, 32) = 2.767, *p* = 0.10; Figure 4) versus cells exposed to pre- and post-placebo therapy plasma both set at 1.0. Interestingly, macrophages exposed to 10% post-placebo- or post-atorvastatin-treated subject plasma both display increased LDL receptor mRNA at 1.40 (95% CI (1.02 to 1.91), F_time_ (1, 32) = 5.299, *p* = 0.03; Figure 4) versus cells exposed to pre-placebo- or pre-atorvastatin-treated subject plasma, each compared to respective pre-treatment plasma set at 1.0.

### 3.5. Statin Therapy Does Not Improve Cholesterol Efflux in Macrophages Exposed to SLE Plasma

Cholesterol efflux capacity: mean efflux from macrophages exposed to subject plasma obtained before placebo therapy was 68.0% (SD = 5.2, n = 3) versus after placebo therapy at 75.8% (SD = 2.6, n = 2; Figure 5A) and mean efflux before statin therapy was 49.8% (SD = 27.7, n = 2) versus after statin therapy at 53.9% (SD = 25.8, n = 2; F_interaction_ (1, 3) = 0.355, *p* = 0.593). This difference was not significant. Intracellular FC levels were reduced in macrophages exposed to 10% plasma from post- either placebo or statin therapies as compared to pre-placebo or statin therapies: mean pre-placebo FC was 6.29 μg/mL (SD = 0.69, n = 2); mean post-placebo FC was 5.28 μg/mL (SD = 0.17, n = 2); mean pre-statin FC was 4.77 μg/mL (SD = 1.31, n = 5); mean post-statin FC was 4.32 μg/mL (SD = 1.00, n = 5; F_time_ (1, 5) = 8.878, *p* = 0.030; Figure 5B).

However, the intracellular CE to FC ratio was not reduced significantly: mean pre-placebo ratio was 1.35 (SD = 0.69, n = 3); mean post-placebo ratio was 1.15 (SD = 0.53, n = 3); mean pre-statin ratio was 1.35 (SD = 0.17, n = 5); mean post-statin ratio was 1.09 (SD = 0.16, n = 5; F_interaction_ (1, 6) = 0.0529, *p* = 0.825).

### 3.6. Statin Therapy Does Not Reduce Oxidized Cholesterol Uptake in Macrophages Exposed to SLE Plasma

Next, we examined oxLDL uptake in macrophages in the presence of SLE plasma. The mean uptake by macrophages exposed to pre-placebo plasma was 521,321FU (SD = 230,000; n = 2) versus after-placebo at 131,027 FU (SD = 96,137; n = 2); mean uptake by macrophages exposed to subject plasma obtained pre-statins was 212,140 FU (SD = 23,741; n = 2) versus post-statins at 170,926 FU (SD = 18,553; n = 2; F_interaction_ (1, 2) = 13.46, *p* = 0.067; Figure 6). This difference was not significant.

## 4. Discussion

Previous work by our group has demonstrated that plasma from patients with SLE impairs cholesterol metabolism in cultured human macrophages [9,11,12]. Here, we explored the modulating effects of placebo and statin therapy on the ability of SLE plasma to alter cholesterol transport in an atherogenic direction in THP-1 human macrophages, a well-accepted model for atherosclerosis [20].

Macrophages play a central role in atherogenesis through the accumulation of cholesterol and the production of inflammatory mediators and cytokines [21]. Lipid-overloaded macrophages, or foam cells, are primary cellular components of atherosclerotic lesions through all stages of development [22]. Total cellular content of cholesterol in macrophages is determined by uptake (mediated by scavenger receptors), outflow of cholesterol (mediated by reverse cholesterol transport proteins), and biosynthesis via the mevalonate pathway [23].

Like any dynamic system, cholesterol movement must be carefully regulated—transport of cholesterol into the cell must be balanced by removal. In this study, we exposed naïve THP-1 macrophages to plasma from LAPS study SLE participants taken before and after initiation of placebo or statin therapy and evaluated change in expression of genes that perform lipid transport functions and serve as a barometer of cholesterol homeostasis. For influx, we measured message for scavenger receptors CD36, ScRA, and LOX-1 as well as the LDL receptor, and for efflux we measured message for 27-hydroxylase, ABCA1, ABCG1, and LXRα. Prominent members of the scavenger receptor family are the class A type I and II receptors (SR-A), and the class B receptors SR-BI and CD36, which, unlike the native LDL receptor, are not feedback controlled. The CD36 cell surface receptor is responsible for high-affinity recognition and internalization of oxLDL by human monocyte-derived macrophages and is a major participant in foam cell formation [24]. ABCA1 mediates active efflux of cholesterol from cells to apolipoproteins; reducing foam cell formation. ABCA1 functions as a rate-controlling protein in the apoA-I–dependent active transport of cholesterol and phospholipids [25]. ABCG1 is critically involved in regulation of lipid-trafficking mechanisms in macrophages and participates in cholesterol and phospholipid efflux [25]. The 27-hydroxylase enzyme constitutes one of the first lines of defense in atherosclerosis prevention. Cholesterol 27-hydroxylase activity provides a pathway for elimination of intracellular cholesterol by conversion to more polar oxysterol metabolites such as 27-hydroxycholesterol [26]. 27-hydroxycholesterol behaves like a statin, potently inhibiting HMG CoA reductase while also suppressing smooth muscle cell proliferation and diminishing macrophage foam cell formation.

27-Hydroxycholesterol is a critical signaling molecule for expression of ABCA1 [27]. LXRs function as sterol sensors by responding to increases in 27-hydroxycholesterol and other oxysterols with upregulated transcription of gene products that control cholesterol catabolism and efflux [27]. Activation of LXRs by endogenous oxysterol ligands, such as 27-hydroxycholesterol, induces transcription of ABCA1 and ABCG1. The importance of cholesterol efflux in atherosclerosis has been validated in studies that show it is inversely associated with cardiovascular events [28,29].

We demonstrate here that SLE plasma-induced disruption of the balance between efflux and influx genes in THP-1 human macrophages is not corrected by statin treatment. Compared to pre-placebo and statin treatment SLE plasma, post-placebo and statin treatment SLE plasma upregulated the LDL receptor and suppressed both ABCA1 and ABCG1 in THP-1 macrophages. Our results are consistent with previous reports of reduced ABCA1 and ABCG1 and increased LDL receptors caused by statins in macrophages [30,31,32], but this change was also observed in macrophages exposed to placebo-treated plasma. Since all LAPS subjects have SLE, the clinical course of this chronic disease entails progression with persistent disease activity over time [33]. Long-term durable remission is rare. Plasma from each patient was collected at two different time points separated by 2 years and therefore the disease duration was 2 years longer in the post-statin or post-placebo later time-point. Whether given statin or placebo, SLE progression leads to persistent inflammation promoting abnormal lipid handling. Similar response in post-placebo and post-statin treatments confirms ineffectiveness of statins to control CVD.

To our knowledge, ours is the first study to confirm that statins given orally to human subjects change the properties of their plasma in a way concordant with its effects when added to macrophages directly. This link may provide a paradigm for future mechanistic CVD drug studies.

SLE plasma exhibits atherogenic potency by attenuating the capacity of macrophages to defend against excess cholesterol and maintain lipid homeostasis [9,12]. Statins not only suppress cholesterol biosynthesis, but also have cholesterol-independent pleiotropic anti-inflammatory effects that led to studies such as LAPS to explore their possible utility in SLE. However, recent findings indicate that cardiovascular benefits are conferred by LDL lowering rather than pleiotropic actions and that statins do not influence SLE disease activity [34,35]. The present LAPS substudy supports these clinical results at a molecular and cellular level as statin treatment did not make the lupus plasma environment less atherogenic nor did it alter efflux capacity. Lack of change in efflux capacity with statins was documented previously by Yamamoto’s group and in persons with type 2 diabetes by Muñoz-Hernandez et al. [36,37]. We also observed no change in PPARγ, although other studies found that statin therapy increased PPARγ receptors and reduced inflammation [38]. In our study, both placebo and statin treatment decreased intracellular free cholesterol and raised 27-hydroxylase. Increased 27-hydroxylase may be a compensatory mechanism since oxysterol production by 27-hydroxylase would upregulate the ABC transporters. We found that the pro-atherogenic milieu in the hyper-inflammatory SLE plasma was little changed with statin treatment in our cell culture system as compared to placebo treatment. This work provides an in vitro mechanistic rationale for the lack of impact on atherosclerotic progression seen in the LAPS study, as well as the pediatric SLE atorvastatin trial [39]. The combined results of LAPS and this sub-study may prompt future evaluation of CVD treatments in SLE subjects that target the cholesterol transport system and improve efflux capacity [40]. Perhaps adding this type of treatment to statins could improve CVD outcomes in lupus patients.

This study has a number of limitations. The design of LAPS did not include a cohort of non-SLE patients on statins, so we have only published data showing lack of statin effect on cholesterol efflux capacity in non-lupus subjects [36,37]. Rosuvastatin administered for two months has been shown to induce PPAR-γ in circulating monocytes of human subjects with coronary artery disease [41], but we did not see this change in our lupus cohort. Due to the small volume of plasma available to us, we were not able to perform Western blots and so rely on mRNA levels. Despite these issues, LAPS provided a unique opportunity to evaluate lupus plasma properties after long-term use of a statin.

## 5. Conclusions

This is a small study using a macrophage cell culture model as a surrogate for the artery wall, but it highlights the need for new avenues of treatment in SLE patients at risk for atherosclerotic cardiovascular disease. Although statins alone may not be optimal cardioprotective drugs in lupus, the fact that they reduce cellular cholesterol burden and show efficacy in treating dyslipidemia in general make them good candidates for combination therapies [42]. Further studies are indicated in order to address the urgent need to reduce cardiovascular risk in the lupus population [43].

## Figures and Tables

**Figure 1 medicina-55-00514-f001:**
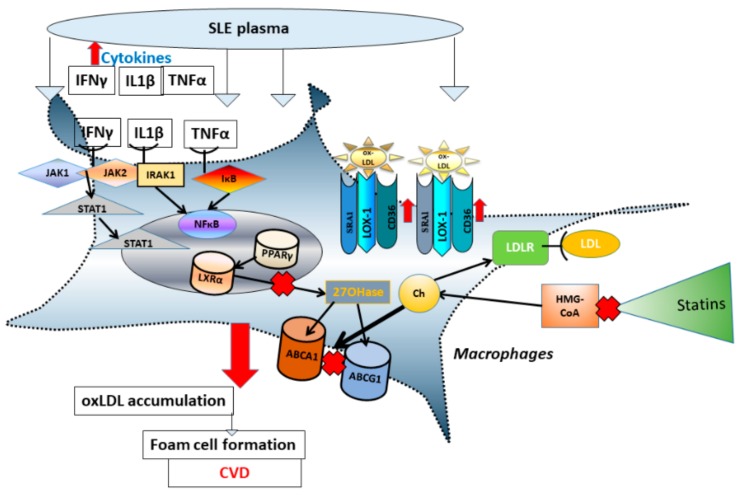
Statins do not compensate for atherogenic properties of SLE plasma and do not correct SLE-induced cholesterol transport abnormalities. Plasma from SLE patients contains elevated levels of major pro-inflammatory cytokines: IFNγ, IL-1β, and TNFα. The inflammatory process in the presence of oxLDL triggers atherosclerotic fatty streak formation in the artery wall. On the cellular level these changes correspond to abnormal oxLDL accumulation and suppression of macrophage cholesterol efflux via ABCA1 and ABCG1 pathways. OxLDL, unlike native LDL, easily enters the cell through scavenger receptors (CD36, SRA1, LOX-1), rather than through the LDLR. Statins inhibit HMG-CoA reductase, which decreases Ch production within the cell and stimulates LDLR synthesis. However, statins are not able to alter pathways affected by inflammation and oxLDL in patients with SLE. Abbreviations: ATP binding cassette (ABC), interferon gamma (IFNγ), cholesterol (Ch), interleukin 1 β (IL-1β), Janus kinase (JAK), LDL receptor (LDLR), liver X receptor (LXR), lectinlike oxidized LDL receptor (LOX), nuclear factor kappa-light-chain-enhancer of activated B cells (NFkB), oxidized LDL (oxLDL) peroxisome proliferator-activated receptor (PPAR), scavenger receptor A1 (ScRA1), Signal Transducer and Activator of Transcription proteins (STATs), tumor necrosis factor α (TNFα), 27-hydroxylase (27OHase).

**Figure 2 medicina-55-00514-f002:**
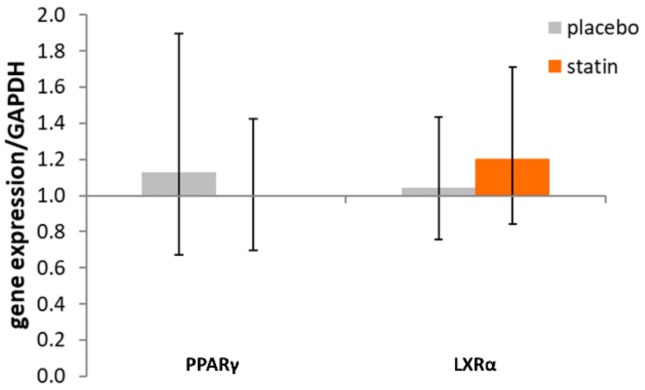
Effect of SLE plasma (post versus pre-placebo and post versus pre-statin) on the expression of nuclear receptors PPARγ and LXRα in THP-1 human macrophages. THP-1 macrophages were incubated for 18 h in the presence of 10% plasma from pairs of before and after placebo or statin treatment, as indicated in the methods. Following incubation, total RNA isolated from cells of each condition was reverse transcribed and amplified by QRT-PCR with GAPDH message as an internal standard. Gene expression levels were graphed as fold changes in mRNA expression compared to before placebo- or statin-treated cells. The data represent fold change ± 95% CI.

**Figure 3 medicina-55-00514-f003:**
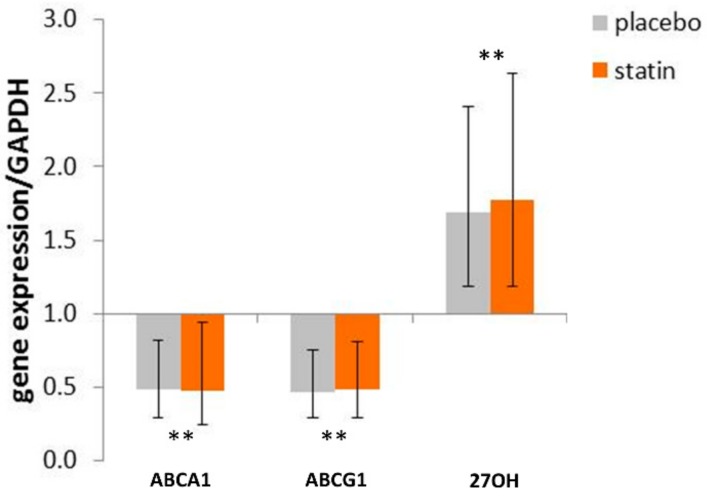
Effect of SLE plasma (post versus pre-placebo and post versus pre-statin) on the expression of cholesterol efflux genes in THP-1 human macrophages. THP-1 macrophages were incubated for 18 h in the presence of 10% plasma from pairs of before and after placebo or statin treatment, as indicated in the methods. Following incubation, total RNA isolated from cells of each condition was reverse transcribed and amplified by QRT-PCR with GAPDH message as an internal standard. Gene expression levels were graphed as fold change in mRNA expression compared to before placebo- or statin-treated cells. The data represent fold change ± 95% CI. ** *p* < 0.01.

**Figure 4 medicina-55-00514-f004:**
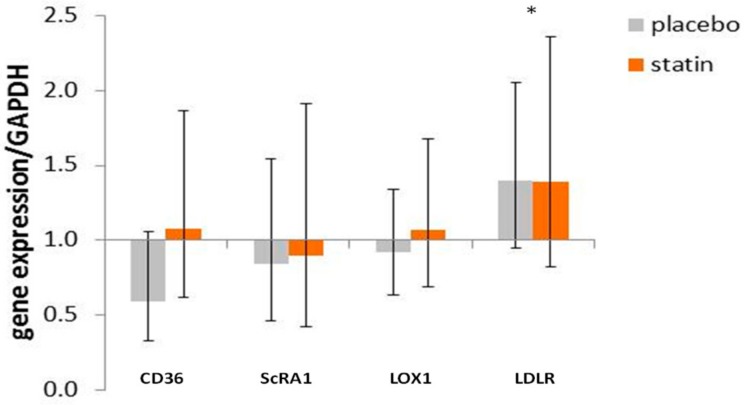
Effect of SLE plasma (post versus pre-placebo and post versus pre-statin) on the expression of cholesterol influx proteins and receptor for LDL in THP-1 human macrophages. THP-1 macrophages were incubated for 18 h in the presence of 10% plasma from pairs of before and after placebo or statin treatment, as indicated in the methods. Following incubation, total RNA isolated from cells of each condition was reverse transcribed and amplified by QRT-PCR with GAPDH message as an internal standard. Gene expression levels were graphed as fold change in mRNA expression compared to before placebo- or statin-treated cells. The data represent fold change ± 95% CI. * *p* < 0.05.

**Figure 5 medicina-55-00514-f005:**
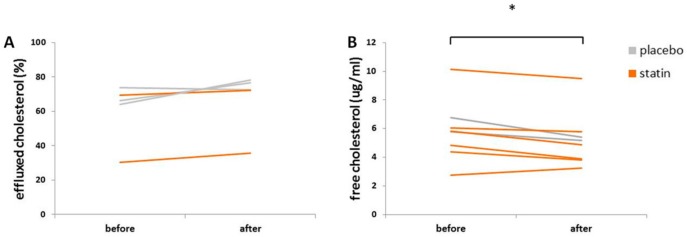
Effect of SLE plasma obtained before and after patient was treated with statin or placebo on cultured macrophage cholesterol efflux and free cholesterol. THP-1 macrophages were incubated for 18 h in the presence of 10% plasma from pairs of before and after placebo or statin treatment, as indicated in the methods and analyzed using the Amplex Red cholesterol assay kit. (**A**) Effluxed cholesterol as a percent of total cholesterol levels and (**B**) intracellular free cholesterol. The data represent pairs of before and after values. * *p* < 0.05.

**Figure 6 medicina-55-00514-f006:**
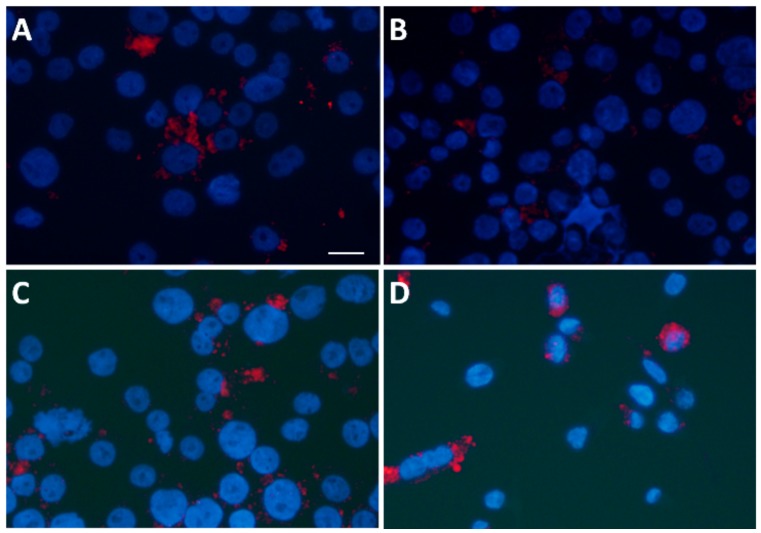
Effect of SLE plasma pre- and post-placebo and pre- and post-statin on oxidized LDL (oxLDL) uptake in THP-1 macrophages. THP-1 macrophages were incubated for 18 h in the presence of 10% plasma from pairs of before and after placebo or statin treatment, as indicated in the methods, and incubated with red fluorescent oxLDL. (**A**) oxLDL uptake pre-placebo treatment, (**B**) oxLDL uptake post-placebo treatment, (**C**) oxLDL uptake pre-statin treatment, and (**D**) oxLDL uptake post-statin treatment. Scale bar = 20 µm; DAPI-stained nuclei are blue.

**Table 1 medicina-55-00514-t001:** Demographic and clinical characteristics at entry.

Group		Placebo	Atorvastatin
n		21	23
gender (%)	female	100	87
	male	0	13
race (%)	African American	48	65
	Caucasian	52	35
age (mean ± SD)		32 ± 12	34 ± 10
disease duration years (median (range))		2 (0–20)	2 (0–22)
anti-ANA antibodies present (%)		100	100
anti-ANA titer (median (range))		1280 (40–20,480)	640 (160–2560)
anti-dsDNA antibodies present (%)		67	74
anti-Sm antibodies present (%)		10	18
RNP antibodies present (%)		30	30
low C3 or C4 (%)		69	67
lupus nephritis (%)		17	32

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
