# Peer review of "Human Lupus Plasma Pro-Atherogenic Effects on Cultured Macrophages Are Not Mitigated by Statin Therapy: A Mechanistic LAPS Substudy"

_medicina, 2019, doi:10.3390/medicina55090514_

Round 1
Reviewer 1 Report
The submitted manuscript rises an interesting topic of atherosclerotic cardiovascular disease in the course of systemic lupus erythematosus. The topic itself might be interesting for the readers of Medicina. Moreover, being a substudy of a LAPS, the standards for the article are set quire high. Unfortunately, the presented manuscript does not reach this standards, mainly due to inadequate and deficient experimental scheme. The first doubt arises at the beginning of the method section, when authors describe differentiation of THP-1 cells into macrophages upon stimulation of PMA. Was the purity of the acquired population evaluated? What method was used to do so? What was the final percentage of macrophages used in the following tests? This uncertainty considering the cells used in the study, makes following experimental steps doubtful as well. Moreover, the evaluation of proteins only on mRNA level is not enough to draw meaningful conclusions. I do understand the limited amount of samples, but Western Blots are not the only method used for the assessment on protein concentration in serum.
Additionally, there is no explanation for the concentration of patients plasma, used in the study. Why it is 10% for cell culture but only 5% for cholesterol efflux analysis?
Finally, I do think that healthy individuals plasma should be used in the study, to whether it can somehow influence the expression of the analysed genes and cholesterol pathways.
Unfortunately, in my opinion, the presented manuscript can not be published in the present for, as it requires major changes both in the experimental line and in the text itself. Therefore I recommend rejection.
Author Response
Response to Comments of Reviewer:
Reviewer 1:
The submitted manuscript raises an interesting topic of atherosclerotic cardiovascular disease in the course of systemic lupus erythematosus. The topic itself might be interesting for the readers of Medicina. Moreover, being a substudy of a LAPS, the standards for the article are set quire high. Unfortunately, the presented manuscript does not reach this standards, mainly due to inadequate and deficient experimental scheme. The first doubt arises at the beginning of the method section, when authors describe differentiation of THP-1 cells into macrophages upon stimulation of PMA. Was the purity of the acquired population evaluated? What method was used to do so? What was the final percentage of macrophages used in the following tests? This uncertainty considering the cells used in the study, makes following experimental steps doubtful as well. Moreover, the evaluation of proteins only on mRNA level is not enough to draw meaningful conclusions. I do understand the limited amount of samples, but Western Blots are not the only method used for the assessment on protein concentration in serum.
RESPONSE: We apologize for omitting detail and have added to the description of our process in the Methods Section. The method of monocyte to macrophage differentiation with PMA stimulation is a traditional method utilized in many laboratories including ours. Cell differentiation, morphology and uniformity of cell adhesion per well was verified by evaluating cell adhesion and spreading under an optical microscope. We confirmed that: 1) cells were equilibrated with growth medium without PMA for 24 hours after macrophage differentiation and attachment, 2) the amount of cells was equal in all experimental wells; 3) treatment was performed in triplicate. This approach minimizes errors in cell culture technique and secures unity of our experiments.
Additionally, there is no explanation for the concentration of patients plasma, used in the study. Why it is 10% for cell culture but only 5% for cholesterol efflux analysis?
RESPONSE: All experiments shown in this manuscript were done at the same 10% concentration of plasma on THP-1 macrophages. The authors regret the error and it is now corrected in the manuscript.
Finally, I do think that healthy individuals plasma should be used in the study, to whether it can somehow influence the expression of the analysed genes and cholesterol pathways.
RESPONSE: Our group has published a number of manuscripts utilizing plasma from patients with autoimmune diseases, such as RA and SLE. There we have described alterations in cholesterol handling in endothelial cells and macrophages initiated by factors present in plasma from these patients in comparison to plasma from healthy controls. This study was design to evaluate changes in the same individual before and after treatment with statins. Since no control group (healthy individuals treated with statins) was used in the LAPS study, comparing pre- and post-statin treatment in SLE subjects was the only approach possible within the LAPS study and is in accord with ethical considerations since statins are not given to healthy persons for no indication.
Reviewer 2 Report

In this manuscript, the authors claim to investigate the biological effect of statin in Atherosclerosis of lupus patients. With this aim in view, they assessed the expression level of molecules (mainly evaluation of the mRNA expression level) involved in cholesterol metabolism in the presence of lupus sera treated with placebo or statin.
Although the question is of interest and the authors use sera from lupus patients to decipher whether statin exerts an effect on macrophages (THP-1 cells), the data should be dramatically improved and the rationale associated with some parts in this manuscript remain difficult to understand. For instance, how the authors explain that the placebo treatment modulated the expression level of their genes (not significant due to elevated SD ?)
In general, the rationale of the study has to be dramatically improved (why authors did not perform transcriptomic analysis of pre- and post-statin exposed macrophages) and this study remains too preliminary to be published.
Author Response
Reviewer 2:
In this manuscript, the authors claim to investigate the biological effect of statin in Atherosclerosis of lupus patients. With this aim in view, they assessed the expression level of molecules (mainly evaluation of the mRNA expression level) involved in cholesterol metabolism in the presence of lupus sera treated with placebo or statin.
Although the question is of interest and the authors use sera from lupus patients to decipher whether statin exerts an effect on macrophages (THP-1 cells), the data should be dramatically improved and the rationale associated with some parts in this manuscript remain difficult to understand.
RESPONSE: We have rearranged the structure of the manuscript, added a scheme, underlining the mechanistic changes in macrophages in SLE not corrected by statins and we hope that this improves the clarity of our presentation. We believe that in the revised format our manuscript will be easier to understand.
For instance, how the authors explain that the placebo treatment modulated the expression level of their genes (not significant due to elevated SD ?)
RESPONSE: In our study the placebo treatment has the same change as statin treatment. Since all subjects have SLE, the natural history of this chronic disease entails progression. The plasma from each patient was collected at 2 different time points 2 years apart and the disease duration is therefore 2 years longer at the post-statin/placebo time-point. In both cases, SLE progression leads to inflammation, persistently elevated pro-atherogenic cytokines and accumulation of lipids in the vessel walls. Similar response in post-placebo and post-statin treatments confirms ineffectiveness of statins to control CVD.
In general, the rationale of the study has to be dramatically improved (why authors did not perform transcriptomic analysis of pre- and post-statin exposed macrophages) and this study remains too preliminary to be published.
RESPONSE: In this study we present a rationale for statin therapy failure in control of CVD in lupus patients. QRT-PCR and cholesterol efflux analysis are still very useful measures of underlying mechanism that are often used to confirm results of transcriptomic analysis which may produce reams of difficult-to-interpret data. Further, the relevant proteins and pathways of cholesterol transport and homeostasis in macrophages are well-characterized and their effects have been documented by us and others. More detailed analysis is planned for the future. LAPS was a landmark study and we believe that our results and proposed mechanism explaining statins ineffectiveness in SLE patients merits publication and will raise awareness in the medical and research community and spur further investigation.
Overall, the structure of the manuscript was carefully re-arranged, and the figures and figure legends updated and clarified.
Round 2
Reviewer 1 Report
Dear Authors,
Thank You for putting so much effort in Your manuscript correction. I am satisfied with Your work.
Congratulations.